Spatial-temporal hypergraph convolutional network for traffic forecasting

Zhao Zhenzhen 1
Shen Guojiang 1
Zhou Junjie 2
Jin Junchen 3
Kong Xiangjie xjkong@ieee.org 1
1 College of Computer Science and Technology, Zhejiang University of Technology , HangZhou , China
2 College of Control Science and Engineering, Zhejiang University , HangZhou , China
3 Zhejiiang Supcon Information Co., LTD , HangZhou , China
Stević Željko
Electronic publication date: 2023 Jul 4
Publication date: 2023
Volume: 9
Electronic Location ID: e1450
Received 2023 Mar 22; Accepted 2023 Jun 1
Copyright: ©2023 Zhao et al.
Copyright year: 2023
Copyright holder: Zhao et al.
License: This is an open access article distributed under the terms of the Creative Commons Attribution License, which permits unrestricted use, distribution, reproduction and adaptation in any medium and for any purpose provided that it is properly attributed. For attribution, the original author(s), title, publication source (PeerJ Computer Science) and either DOI or URL of the article must be cited.
License URL: https://creativecommons.org/licenses/by/4.0/

Keywords: Spatial-temporal dependencies, Hypergraph convolutional network, Traffic forecasting

Funding: “Pioneer” and “Leading Goose” R & D Program of Zhejiang 2022C01050 The National Natural Science Foundation of China 62073295 62072409 The Zhejiang Provincial Natural Science Foundation LR21F020003 This work was supported by the “Pioneer” and “Leading Goose” R & D Program of Zhejiang under Grant 2022C01050, by the National Natural Science Foundation of China under Grant 62073295 and Grant 62072409, and by the Zhejiang Provincial Natural Science Foundation under Grant LR21F020003. The funders had no role in study design, data collection and analysis, decision to publish, or preparation of the manuscript.

==============================
Accurate traffic forecasting plays a critical role in the construction of intelligent transportation systems. However, due to the across road-network isomorphism in the spatial dimension and the periodic drift in the temporal dimension, existing traffic forecasting methods cannot satisfy the intricate spatial-temporal characteristics well. In this article, a spatial-temporal hypergraph convolutional network for traffic forecasting (ST-HCN) is proposed to tackle the problems mentioned above. Specifically, the proposed framework applies the K-means clustering algorithm and the connection characteristics of the physical road network itself to unify the local correlation and across road-network isomorphism. Then, a dual-channel hypergraph convolution to capture high-order spatial relationships in traffic data is established. Furthermore, the proposed framework utilizes a long short-term memory network with a convolution module (ConvLSTM) to deal with the periodic drift problem. Finally, the experiments in the real world demonstrate that the proposed framework outperforms the state-of-the-art baselines.

Introduction

With the continuous development of the economy, the car ownership of urban residents is increasing year-by-year and traffic congestion is becoming more and more serious. Traffic congestion is prone to traffic accidents, and traffic accidents restrict the sustainable development of cities (Nagy & Simon, 2018). Meanwhile, the intelligent transportation system (ITS) can apply advanced edge computing technology (Kong et al., 2022b) and control strategy to integrate urban resources, thereby alleviating the contradiction between people, vehicles, and roads, which is of great significance to urban traffic management (Balasubramanian et al., 2023). Therefore, the construction of intelligent transportation systems is imminent. Traffic prediction is the easiest and most intuitive way to perceive changes in urban road conditions. Based on the city road state, the urban traffic managers can control the signal lights in the road to increase the effective capacity of urban roads and enhance the travel experience of urban residents. Luckily, with the continuous maturity of industrial technology, road sensors can effectively provide high-quality road information, bringing new opportunities to solve the problem of traffic forecasting. Traffic forecasting is the process of analyzing historical road information, such as traffic flow, speed, occupancy, etc., to predict the trend of road changes. As we all know, traffic prediction is a typical spatial–temporal modeling problem. Especially after the emergence of neural networks, many scholars apply the powerful feature extraction capabilities of deep neural networks to research traffic prediction problems from spatial and temporal dimensions. To capture the temporal trend, recurrent neural network (RNN) and its variants, such as long short-term memory network (LSTM) (Ma et al., 2015), are applied to it. Moreover, to model the spatial relationship, convolutional neural networks (CNN) (Liu et al., 2017) are also widely used. Inspired of graph convolution networks (GCN) (Kipf & Welling, 2017), extensive researches (Yu, Yin & Zhu, 2018; Wu et al., 2019; Yao et al., 2019; Zhao et al., 2019; Guo et al., 2019; Wang et al., 2020; Zheng et al., 2020; Bai et al., 2020; Oreshkin et al., 2021; Zhu et al., 2022; Kong et al., 2022a) had been carried out on how to model spatial–temporal graph in traffic prediction. These jobs make much progress and motivate us a lot. However, the main disadvantage of these works is that they conduct simple graphs to describe the relationship between pairs of nodes. Thus, the analysis of multiple key traffic intersections across the road network in the spatial dimension was problematic. Therefore, how to construct the hypergraph based on traffic information and extract high-level features is still an area worth exploring.

To intuitively illustrate the across road-network isomorphism and the periodic drift, this article present a schematic figure to elaborate. As is shown in Figs. 1A, 1C, pairwise analysis of the influence of traffic node 4 on other traffic nodes using a simple graph is not appropriate. This is because the road information in the important traffic sensor 4 can affect the information on other traffic sensors that are not physically directly connected to it. Although different nodes are not physically adjacent, the nodes are homogeneous across the road network. While the hypergraph can treat multiple nodes as a hyper point so that it can analyze the higher-order relationship between multiple nodes. By doing so, the spatial features hidden in the traffic data can be fully excavated. Meanwhile, the time trend does not strictly follow the periodicity, it has a certain drift. This article visualizes the traffic speed on a certain road node from 4:00 am to 10:00 am on weekdays. As is shown in Fig. 1B, what can be seen is that the peak traffic speed during the weekday will be reached at different time steps. Therefore, when capturing time dependencies, it is difficult to deal with the drift without increasing the perception field to extract time trends from other nodes.

Figure 1 Complex spatio-temporal correlations.

(A) The sensor distribution on the road network. (B) The periodic drift in temporal dimension on different days. (C) The spatial relationships among multiple road nodes and the temporal trend change along time.

To the best of our knowledge, this article propose a spatial–temporal hypergraph convolutional network for traffic forecasting (ST-HCN). Firstly, the proposed network applies the K-means clustering algorithm to find the key intersections in the traffic data from the global level. Secondly,the proposed network utilizes the local connection relationship to construct the hypergraph to represent the local correlation and across road-network isomorphism. Thirdly, a dual-channel hypergraph convolution to integrate the features of the super edge with the features of the node is designed, exploring the high-order spatial relationships of traffic data. Finally, the proposed network adopted a long short-term memory (LSTM) network with a convolution module, discovering the characteristics of time series from multiple traffic nodes to tackle the periodic drift problem. Our contributions can be summarized as follows:

• This article proposed a hypergraph construction method for traffic data. It can model the spatial relationship from global to local, discovering the complex across road-network isomorphism hidden in the traffic data. Hypergraph provides a natural way to capture beyond-pairwise relations, it can directly represent the high-order spatial relation among more than two nodes.

• This article proposed a dual-channel hypergraph convolution method. The hypergraph explores the high-order relationship between nodes and the line graph explores the relationship between hyperedges. The fusion of line graph and hypergraph convolution can fully characterize the many-to-many spatial relationship in the road network.

• This article evaluated the proposed methods on the datasets from the real world, and the experimental results demonstrate the superiority of the proposed methods.

In the following, this article firstly introduces the related work about traffic prediction and convolution on graphs in Section 2. Secondly, this article presents some preliminary concepts and an overview of our framework ST-HCN in Section 3. Thirdly, this article introduces the proposed methods in detail in Section 4. Fourthly, this article describes our experiment settings and verify the effectiveness of the proposed framework in Section 5. At last, this article makes a conclusion in Section 6.

Literary Review

The most important thing in traffic forecasting is the spatiotemporal modeling of traffic data. This article is supposed to solve the across road-network isomorphism in the spatial dimension and the periodic drift in the temporal dimension. Thus, this article will review the related work about traffic prediction and convolution on graphs, hoping to be inspired by previous work.

Traffic prediction

To date, significant achievements have been made in traffic forecasting. These methods can be divided into three categories: statistical methods, machine-learning-based methods, and deep-learning-based methods. Statistical methods, such as historical average (HA), auto-regressive integrated moving average (ARIMA) (Williams & Hoel, 2003), and vector auto-regressive (VAR) (Zivot & Wang, 2006). The statistical methods predict well in certain assumptions. However, the dynamic temporal features of traffic data can not satisfy the assumptions. Therefore, some traditional machine learning methods are proposed to cope with traffic prediction problems. Machine-learning based methods, such as support vector regression (SVR) (Chen et al., 2015) and random forest regression (RFR) (Jo-hansson et al., 2014). Nevertheless, machine-learning methods are problematic in dealing with dynamic changes in data. Thanks to the powerful feature extraction capability of deep learning, many researchers have modeled both spatiotemporal features from a data-driven perspective to achieve accurate traffic prediction. Ma et al. (2015) and Liu et al. (2017) apply the LSTM and CNN into the traffic prediction. Nevertheless, CNN cannot adequately model the road network. Therefore, with the advent of GCN, researchers had drawn attention to understanding the pattern of the spatial–temporal graph. Yu, Yin & Zhu (2018) combined the GCN and the gated convolutional neural network to model the traffic data, enabling faster training speed with fewer parameters. Zhao et al. (2019) proposed a temporal graph convolutional network (T-GCN) model, which combines GCN and gated recurrent unit (GRU). GCNs are used to learn complex topologies to capture spatial dependencies, while GRUs are used to learn the dynamics of traffic data to capture temporal dependencies. However, the pre-defined simple graphs have limits in reflecting the dynamics of traffic data. Wu et al. (2019) developed a novel adaptive dependency matrix and learn it through node embedding, which can precisely capture the hidden spatial dependency in the data. Zheng et al. (2020) designed a graph multi-attention network(GMAN), in which an attention conversion module is included between the encoder and decoder to simulate the relationship between historical time steps and future time steps, helping to alleviate the problem of error propagation between prediction time steps on the road network graph. Bai et al. (2020) designed a data-adaptive graph generation module to infer the inter-dependencies among different traffic series automatically. Huang et al. (2021) augmented the original road network into a region-augmented network, in which the hierarchical regional structure can be modeled. Yu et al. (2021) proposed a novel deep spatio-temporal graph convolutional network, learning the spatial correlations, temporal dynamic interactions and external influences in traffic-relevant heterogeneous data, for traffic accident prediction. Oreshkin et al. (2021) achieved performance competitive with or better than the best existing algorithms, without requiring knowledge of the graph. These researchers made great achievements in applying GCN to the field of traffic prediction and motivated us a lot. However, their methods are still based on simple graphs for data modeling. Simple graphs are insufficient for capturing the complex spatiotemporal characteristics of traffic data. Although they can illustrate the relationship between paired nodes, they fail to depict the high-order relationship between many-to-many nodes. Consequently, the depiction of spatiotemporal features is incomplete, hindering the achievement of the desired outcome. Thus, how to extract the many-to-many spatial relationship hidden in the traffic data with a hypergraph is worth exploring.

Convolution on graphs

The powerful feature extraction capability of CNN is largely attributed to the existence of convolution kernels, which can well extract features from data and make use of them. Motivated by the idea of convolution kernel, many scholars pay attention to applying the convolution operation to graph-structured data, which results in the so-called GCN. Spectral-based GCN owns a solid mathematical foundation in the field of signal processing (Shuman et al., 2013), where the graph is assumed to be undirected and the Fourier transformation is applied to convert the convolution operation into product operation. For an undirected graph, it is evident that the normalized graph Laplacian matrix L is symmetric and positive semi-definite. The normalized graph Laplacian matrix can be factored as L = UΛUT by applying eigenvalue decomposition. Nevertheless, the computational complexity is high. Therefore, Kipf & Welling (2017) firstly introduced the fast-approximate convolution on the graph with layer-wise propagation rule for semi-supervised node classification. On the other hand, it’s unsuitable for the traffic forecasting problem because the proposed GCN depends on the static adjacency matrix consisting of 0 or 1. Thus, many scholars try to construct the meaningful adjacency matrix to fit the real road network. Yu, Yin & Zhu (2018) and Zhao et al. (2019) applied Gaussian kernel function to create a weighted adjacency matrix for graph convolution. Guo et al. (2019) combined the GCN and attention mechanism to design the dynamic adjacency matrix to catch the dynamic spatial information of the road network. However, since GCN must require modeling the data as an undirected graph, this will result in the loss of directional modeling of the data. Therefore, Han et al. (2020) designed a dirgraph convolutional neural network (DGCN)-based learning model to tackle the congestion recognition problem. Shen et al. (2022) proposed an attention mechanism-based digraph convolution network (ADGCN), which incorporates spatiotemporal traffic information with the three-dimensional urban network and partially decouples the global network topology to a single-knot digraph. To achieve convolution on the hypergraph, an easier way to implement is to extend the Laplacian matrix of simple graphs to hypergraphs (Bolla, 1993; Zhou, Huang & Schölkopf, 2006). Based on this idea, Yadati et al. (2019) proposed a new method of training a GCN on the hypergraph. Fu et al. (2019) utilized hypergraph p-Laplacian to preserve the local geometry of samples and then propose an effective variant of GCN. Feng et al. (2019) presented a hypergraph neural networks (HGNN) framework for data representation learning, which can encode high-order data correlation in a hypergraph structure. Furthermore, Jiang et al. (2019) proposed a dynamic hypergraph neural networks framework to tackle the issue that the hidden and important relations are not directly represented in the inherent structure. Bandyopadhyay, Das & Murty (2020) developed the line hypergraph convolutional networks. Xia et al. (2021) combined the line graph and hypergraph to capture the complex high-order information among items in a session-based recommendation. These hypergraph learning methods inspired us a lot. Studies have conclusively demonstrated the effectiveness of hypergraph learning in identifying many-to-many relationships, making it an ideal choice for recommendation systems. Due to the influence of human behavior on traffic data, it often displays cross-road network characteristics in terms of its spatial dimension. This underlines the importance of utilizing the hypergraph construction method when modeling traffic spatiotemporal data. Therefore, how to construct a hypergraph according to the characteristics of traffic data and the potential of hypergraph for traffic forecasting has remained unexplored.

Overview

In this section, this article introduces some notations, definitions, concepts, and an overview of the proposed framework.

Preliminary

In this article, a novel traffic information forecasting framework is proposed, which can be traffic flow, traffic speed, or traffic density, on the roads. Without loss of generality, this article uses traffic speed as an instance of traffic information to verify the effectiveness of the proposed framework.

Definition 3.1 (Road network). A road network represents the topological structure of the physical road network, which can be described as G=V,E,W.V is a set of road nodes, V=v1,…,vN, N is the number of road nodes. E is a set of edges. W is the weighted adjacency matrix that is used to represent the connection between roads.

Definition 3.2 (Hypergraph). Let HG=VH,EH denote a hypergraph, where VH is a finite set containing N vertices and EH is a finite set containing M hyperedges. Each hyperedge e ∈ E is given a nonnegative weight we and all the weights formulate a diagonal matrix WH ∈ RM×M. The structure of the hypergraph can be described by a correlation matrix H ∈ RN×M, where the hv,e=1 if the hyperedge e contains a vertex v, otherwise 0. For each vertex v ∈ VH and for each hyperedge e ∈ EH, their degree can be defined as dv=∑e∈EHwehv,e and de=∑v∈VHhv,e.Dv denotes the diagonal matrix of each vertex and De denotes the diagonal matrix of each hypergraph.

Definition 3.3 (Line graph of hypergraph). Given a hypergraph HG=VH,EH, the line graph of hypergraph LG=VL,EL,WL is a graph where each node of LG is a hyperedge in HG and two nodes of LG are connected if the their corresponding hyperedges in HG share at least one common vertex (Whitney, 1992). For each two nodes ei and ej in LG, the weight matrix WLij=ei∩ej/ei∪ej.

Definition 3.4 (Feature matrix). A feature matrix represents the traffic speed of all the road nodes in a certain period of time, which can be described as XTh×N.Th is the time slice of the past period. h denotes the length of the historical time series, X = {xt1, xt2, …, xth}. And xti ∈ Ri×N is the speed of all the road node at time slot i.

Problem Definition Given a road network (G), feature matrix (XTh×N). This article aims at finding the mapping function (f), which can learn the spatial–temporal features from the historical traffic speed, to predict the next Tp time slot, xt+1,xt+2,…xt+p=fG;XTh×N.

Framework

This article proposes a spatial–temporal hypergraph convolutional network for traffic forecasting (ST-HCN). The architecture of ST-HCN is shown in Fig. 2, ST-HCN consists of a hypergraph convolution layer and an LSTM network layer. For a given road network G and feature matrix X, the proposed network first uses the K-means clustering algorithm to construct the hypergraph and the line graph. Secondly, the proposed network exchanges the node feature information extracted by the hypergraph convolution with the hyperedge feature information extracted by the line graph convolution to explore local correlation and across road-network isomorphism in the spatial dimension. Thirdly, the LSTM network with the convolution module is applied to model the traffic data, embedding high-order spatial relationship information in time series to deal with the periodic drift in the temporal dimension. Finally, the extracted spatial–temporal features are fused to make accurate traffic predictions.

Figure 2 Architecture of the proposed spatial–temporal hypergraph convolutional network.

Methodology

In this section, this article elaborates in detail on spatial feature extraction and temporal feature extraction of the proposed framework shown in Fig. 2.

Hypergraph for spatial features

Capturing the spatial features in traffic data is an important problem in traffic prediction. Existing simple graph modeling methods are problematic in capturing the across road-network isomorphism. Therefore, this article introduces the concept of hypergraphs in spatial feature extraction. In the process, it can be mainly divided into two parts, how to construct the hypergraph and how to extract features on the hypergraph. The pseudo-code of spatial feature extraction is shown in Algorithm 1. As is shown in Fig. 3, the proposed framework designed a layer-wised hypergraph convolutional neural network with a residual network structure. This framework feeded the feature matrix X into the two hidden layers. Each hidden layer performs hypergraph convolution and line graph convolution and then passes through the nonlinear activation function Relu. Finally, the embedding vector of the spatial features Z can be obtained.

Figure 3 Overview of spatial feature extraction.

Hypergraph construction

To adaptively find the nodes related to the data distribution in the traffic data, we choose the K-Means clustering algorithm as an implementation method. Firstly,the feature matrix was divided by day and then use the mean value of traffic speed on weekdays to represent the traffic data distribution of different road nodes. Secondly, the traffic speed matrix is fed into the K-Means clustering algorithm and expect the algorithm adaptively to find the cluster center. Thirdly, the road nodes related to the cluster center are aggregated together. By doing so, different clusters selected from the global time distribution trend can be obtained. Furthermore, to capture the local adjacency relationship, the road nodes directly connected to the cluster center node are also added to the cluster. Finally, the proposed framework treated each clustering result as a hyperedge and then complete the construction of the hypergraph.

To capture the across road-network isomorphism in the spatial dimension, this article first applies the K-means clustering algorithm to find the highly correlated traffic nodes globally and utilize the clustering results to construct hyperedges. Furthermore, the cluster center is the most representative node among traffic nodes, which can be regarded as the key traffic intersection, and then the proposed framework uses the physical connection relationship of the road network itself to expand the hyperedges from the cluster center. By doing so, the hypergraph can unify the isomorphism of the across road-network and local correlation in the spatial dimension, characterizing the higher-order spatial relationships beyond paired nodes better. In addition, to denote the relationship between key traffic intersections,a hypergraph-based line graph was constructed. Finally, the combination of the line graph convolution and the hypergraph convolution can fully express the spatial relationship of the traffic data.

Dual channel hypergraph convolution

To achieve convolution on the hypergraph and extract the relationship between the many-to-many nodes in the hyperedge, one of the biggest challenges is how to extend the Laplacian matrix of the simple graph to the hypergraph. Referring to the spectral hypergraph convolution proposed in Zhou, Huang & Schölkopf (2006), the hypergraph Laplacian matrix Δ can be expressed as: (1) Δ=I−Dv−1/2HWHDe−1HTDv−1/2

where I denotes the identity matrix. Following the idea of layer-wise graph convolution (Kipf & Welling, 2017), the formula of hypergraph convolution can be defined as: (2) Xhl+1=ΔXhlPl+bPl

where Pl and bl are the learnable parameter matrix of layer l. The number of nodes in the line graph is the same as the number of hyperedges in the hypergraph. Therefore, to achieve the goal of fusing the influence of key traffic nodes while extracting the many-to-many relationship of nodes, it is necessary to associate the nodes in the line graph with the nodes in the hypergraph. Thus, this article designed a self-learning attention matrix WA ∈ RN×M to learn the mapping relationship between hyperedges and road nodes. Therefore, the line graph convolution can be expressed as: (3) Θhl+1=LsysWAΘhlQl+bQl

where Lsys=D ˆL−1/2A ˆLD ˆL−1/2, A ˆL=WL+I denote the weight matrix of line graph with self-loop, D ˆL is a diagonal matrix, D ˆL=∑jA ˆLij, Ql and bl are the learnable parameter matrix of layer l.

_______________________  Algorithm 1: Dual Channel Hypergraph Convolution Algorithm      _________     Input: Road Networks G, Feature Matrix X;     Output: Spatial Embedding Features Z;   1  Initialize Spatial Embedding Features Z = ∅  2  Initialize average matrix Av = ∅  3  Initialize the parameter K in K-Means Algorithm   4  Split the feature matrix X by day and get the sum of days Ndays   5  for each day data X_d in X do      6   Av = Av + X_d  7  end  8  Av = (1 / Ndays) * Av   9  Cluster centers, Clusters = K-Means.Fit(Av) 10  Take the road nodes closest to the Cluster centers as the key road       nodes 11  for center C in Cluster centers do      12   for Road Node v in road nodes set V do     13   if (C is directly connected with the node v) and (v no in Clusters) then 14   Clusters.append(v) 15   else 16   Continue 17   end 18   end 19  end 20  Get the Hypergraph H(G) for the road network from Clusters. 21  Calculate the Laplacian matrix for H(G) 22  Construct the Line Graph L(G) according to H(G) 23  Hz = Hypergraph Convolutional Layer One (X) 24  Lz = Line Graph Convolutional Layer One (X) 25  Hz = Hypergraph Convolutional Layer Two (Lz) 26  Lz = Line Graph Convolutional Layer Two (Hz) 27  Z = Linear Layer (Concat(Hz:Lz))

Finally, a two-layer dual-channel hypergraph convolution to embed the spatial characteristics of the traffic data was designed. After the feature matrix X passes through the first layer of hypergraph convolution and line graph convolution, the embedding vector of the same dimension can be obtained. And then the first layer of line/hypergraph convolution result is input to the second layer of hypergraph/line graph convolution, which can be expressed as: (4) ConcatLsysWAXh2Q2+bQ2,ΔΘh2P2+bP2

where Θh2=LsysWAXQ1+bQ1, Xh2=ΔXP1+bP1. And by exchanging the information learned by the two convolutions, the dual-channel hypergraph convolution can extract the many-to-many relationship of nodes while obtaining the influence between key traffic nodes.

LSTM network for temporal features

Tackling the periodic drift problem in the temporal dimension is another crucial problem in traffic forecasting. The main reason for the periodic drift is that the time series data of traffic is easily affected by the time series data of other traffic nodes. When capturing the time trend of traffic nodes, the periodic drift shown in Fig. 1B occurs if only the time series of one node is processed. Therefore, this article adopts LSTM with the CNN module (ConvLSTM) to increase the model’s perceptual field to capture information from the time-series information from other road nodes. This temporal dynamic can be captured in this way, alleviating the period drift problem.

As is shown in Fig. 4, the time series data was fed into the convolution based LSTM network, the data of each time node enters the ConvLSTM cell. Then the cell updates the state of the forget gate, memory gate, and output gate through the two-dimensional convolution layer. Finally, the cell output the hidden vector and output vector of the next moment.

The dual-channel hypergraph convolution can obtain the complete spatial information. Nevertheless, the traditional LSTM can only process the time series variables of a single node. Therefore, directly inputting the spatial embedding vector into the LSTM will lose the spatial relationship. The article for precipitation nowcasting (Shi et al., 2015) inspired us a lot. CNN can increase the perceptual field of view through the convolution kernel and then capture local spatial information. Embedding CNN in LSTM enables LSTM to consider the timing information of multiple nodes at the same time, which can be expressed as: (5) it=σWxi∗Xt+Whi∗Ht−1+Wci∘Ct−1+bift=σWxf∗Xt+Whf∗Ht−1+Wcf∘Ct−1+bfot=σWxo∗Xt+Who∗Ht−1+Wco∘Ct−1+boCt=ft∘Ct−1+ΨΨ=it∘tanhWxc∗Xt+Whc∗Ht−1+bcHt=ot∘tanhCt

where ∘ denotes the Hadamard product, ∗ denotes the convolution, σ denotes the sigmoid activation function. The inputs X1⋯Xt, cell outputs C1⋯Ct, and hidden states H1⋯Ht are the 3D tensors, which can directly perform convolution operations. Wh, Wx, WC, and b are the learnable parameter of each layer.

By introducing the convolution method when updating the state of the forget gate, memory gate, and output gate, the perceptual field of view of LSTM is increased. Then, the LSTM can deal with the periodic drift problem in the temporal dimension. Then, the proposed framework regarded the spatial features extracted from the hypergraph as the input of the convolutional LSTM, hoping to further extract temporal features. Finally, the L2 loss is used to measure the performance of our model. The loss function of ST-HCN can be denoted as: (6) loss=1N∑i=1NYi−Y ˆi2

where N represents the number of roads, Yi and Y ˆi denote the ground truth and the model’s prediction respectively.

Figure 4 Overview of temporal feature extraction.

Experiments

In this section, this article will elaborate on the datasets and details used in the experiments in this article. The obtained experimental results are further analyzed from parameter sensitivity, ablation experiment, case study, and analysis of research results and discussion of four aspects. The validity of the proposed model is finally verified.

Data description

This article verified the effectiveness of the proposed model on two real-world traffic datasets, collected by the California Department of Transportation (Caltrans) Performance Measurement System (PeMS), as detailed below. Traffic speed is obtained by calculating the average speed of all vehicles passing the road within a period, and the speed of vehicles passing the road is collected by sensors placed on the road.

PEMSBAY It contains the traffic speed on 325 road sensors in the Bay Area. In addition, the readings of all sensors deployed on the road are also aggregated into 5-minutes windows. This article uses the data in the weekdays from March 6, 2017, to May 5, 2017.

PEMSM It contains the traffic speed on 228 road sensors in the District 7 of California. Furthermore, the readings of all sensors deployed on the road are aggregated into 5-minutes windows. This article uses the data in the weekdays from May 1, 2012, to June 30, 2012.

Experimental settings

In this subsection, this article will introduce the experimental hardware and software environment, model evaluation metric, hyper-parameter settings, and some baseline models. ∅

Hardware and software

The experiments are conducted on a computer with 256 GB memory, an Intel Xeon Gold /2.1 GHz CPU, and a Quadro P6000/24G GPU. Moreover, the proposed approach and all neural network-based baseline models are implemented based on PyTorch 1.7.1 with the cuda 11.0 using the Python language 3.6.10.

Evaluation metric

In the experiments, this article selected the first 35 days (weekdays from March 6, 2017, to April 21, 2017) of historical speed records as the training set, and the remaining as validation and test set respectively for the PEMSBAY dataset. This article selected the first 34 days (weekdays from May 1, 2012, to June 15, 2012) of historical speed records as the training set, and the remaining as validation and test set respectively for the PEMSM dataset. The traffic forecasting problem is a classical regression problem. Thus, to evaluate the prediction performance of different methods, the mean absolute error (MAE), mean absolute percentage error (MAPE) and root mean squared error (RMSE) were selected as the metrics. For the MAE, RMSE, and MAPE metrics, the smaller values indicate the better prediction performance.

• MAE: the average of the absolute errors, which can be defined as: (7) MAE=1N∑i=1NYi−Y ˆi

• MAPE: the percentage of errors to ground truth values, which can be defined as: (8) MAPE=100%N∑i=1NYi−Y ˆiYi

• RMSE: the rooted average squared difference between the predicted values and the ground truth, which can be defined as: (9) RMSE=1N∑i=1NYi−Y ˆi2

Hyper-parameters settings

For the general setting, the hidden size and output size of dual-channel hypergraph convolution are 64 and 8 respectively, the learning rate is 0.001, the batch size is 50, the embedding size of ConvLSTM is 64, the kernel size is 3, the number of layers is 1. For the determination of the number of clusters in the K-Means clustering algorithm, this article makes adjustments according to different datasets based on experimental results. For all the machine learning methods, 70% of the datasets are used for training, and the rest is used for testing. This article uses two layers of dual-channel hypergraph convolution and each layer uses residual connections to ensure the transfer of features.

Baselines

To verify the validity of the proposed methods, this article investigates the classical methods, including historical average (HA) and several common machine learning algorithms, including support vector regression (SVR), K neighbors regression (KNN) and random forest regression (RF). Furthermore, this article compares the proposed methods with representative methods based on GCN and RNN, such as STGCN (Yu, Yin & Zhu, 2018), AGCRN (Bai et al., 2020), and FC-GAGA (Oreshkin et al., 2021). Moreover, to prove the effectiveness of the dual-channel hypergraph convolution module, This article designed three ablation experiments, GST-HCN, HST-HCN, and LST-HCN.

• HA: Historical average uses the average value in the previous periods as the prediction for the future periods.

• SVR: Support vector regression uses a support vector machine to do regression on the traffic sequence.

• KNN: K neighbors finds the k nearest neighbors of a sample and then assigns the average value of certain attributes of these neighbors to the sample.

• RF: Random forest is an ensemble technique that combines multiple decision trees. A random forest usually has a better generalization performance than an individual tree due to randomness.

• STGCN: Spatial-temporal GCN formulates the traffic forecasting problem on graphs and builds the model with complete convolutional structures.

• FC-GAGA: FC-GAGA uses the learnable fully connected hard graph gating mechanism to achieve performance competitive with or better than the best existing algorithms, without requiring knowledge of the graph.

• AGCRN: AGCRN applies the data-adaptive graph generation (DAGG) module to infer the inter-dependencies among different traffic to capture fine-grained spatial and temporal correlations in traffic series automatically based on the two modules and recurrent networks series automatically.

• GST-HCN is a variant of our proposed model, where the dual-channel hypergraph convolution module is replaced by the graph convolution module.

• HST-HCN is a variant of our proposed model, where the dual-channel hypergraph convolution module is replaced by the hypergraph convolution module.

• LST-HCN is a variant of our proposed model, where the dual-channel hypergraph convolution module is replaced by the line graph convolution module.

Experiment results

The results are shown in Table 1 on the datasets PEMABAY and PEMSM. All the tests use 60 min as the historical time window to forecast traffic conditions in the next 10, 15, and 20 min. In other words, this article focuses on short-term traffic forecasting and we highlight the best performance for each forecasting step. Analyzing the experiment results in Tables 1 and 2, the following conclusions can be drawn.

Table 1 Results of different methods on the dataset PEMSBAY.

Classification	Methods	PEMSBAY (10/15/20 min)	
		MAE	MAPE (%)	RMSE	
Classical methods	HA	2.41/2.48/2.56	5.62/5.78/5.97	5.37/5.58/5.76	
SVR	1.43/1.60/1.74	3.46/3.87/4.26	3.23/3.60/3.94	
RF	1.35/1.54/1.72	2.90/3.42/3.90	2.69/3.15/3.56	
KNN	1.45/1.62/1.78	3.20/3.65/4.08	2.92/3.34/3.72	
State-of-the-Art methods	STGCN	1.39/1.68/2.02	3.24/4.02/5.11	2.50/3.10/3.67	
FC-GAGA	1.36/1.65/1.83	3.08/3.91/4.44	2.69/3.45/3.97	
AGCRN	1.37/1.47/1.58	3.23/3.50/3.76	2.77/3.03/3.27	
Our methods	GST-HCN	1.32/1.51/1.68	3.17/3.71/4.21	2.41/2.82/3.18	
HST-HCN	1.41/1.60/1.76	3.68/4.18/4.64	2.61/3.01/3.31	
LST-HCN	1.25/1.46/1.64	2.81/3.45/3.96	2.27/2.73/3.11	
ST-HCN	1.22/1.40/1.57	2.73/3.23/3.73	2.19/2.58/2.94	
Notes.

The best experimental results for each setting are bolded.

Table 2 Results of different methods on the dataset PEMSM.

Classification	Methods	PEMSM (10/15/20 min)	
		MAE	MAPE (%)	RMSE	
Classical methods	HA	3.14/3.22/3.32	7.60/7.81/8.03	6.09/6.29/6.49	
SVR	1.78/2.01/2.21	4.45/5.03/5.54	3.44/3.92/4.36	
RF	1.82/2.10/2.34	4.24/4.95/5.60	3.19/3.76/4.25	
KNN	1.96/2.21/2.43	4.64/5.28/5.87	3.48/3.99/4.44	
State-of-the-Art methods	STGCN	1.87/2.29/2.60	4.32/5.38/6.19	3.16/3.93/4.46	
FC-GAGA	1.86/2.20/2.45	4.36/5.26/5.98	3.38/4.16/4.77	
AGCRN	1.82/1.96/2.09	4.39/4.75/5.08	3.28/3.60/3.89	
Our methods	GST-HCN	1.74/2.00/2.21	4.15/4.79/5.37	2.94/3.41/3.82	
HST-HCN	1.72/1.97/2.19	4.07/4.74/5.29	2.90/3.39/3.75	
LST-HCN	1.68/1.96/2.18	3.87/4.56/5.22	2.88/3.40/3.82	
ST-HCN	1.62/1.86/2.08	3.72/4.34/4.98	2.77/3.23/3.63	
Notes.

The best experimental results for each setting are bolded.

• The HA method is carried out using the historical average value as the prediction result. It can achieve better results on datasets with flat data distribution. However, the analysis of non-linear changes in traffic data is problematic. Therefore, the performance on these two datasets is not ideal. The methods based on machine learning offer an effective way of capturing the nonlinear changes in time series data, having a great improvement than the HA method. In addition, the RF algorithm performs best among machine-learning-based methods. One possible implication of this is that the ability of the RF algorithm can be improved by increasing the number of trees in the forests. Meanwhile, machine learning methods are not inferior to some spatiotemporal graph convolution methods on short-term traffic forecasting. This is largely because the dataset used in this article is recorded on highways in the United States, where the data is relatively smooth and usually does not exhibit non-linear changes. Thus, traditional machine learning algorithms are also a direction worth considering when you want to achieve higher efficiency and not bad precision in short-term traffic forecasting.

• STGCN and FC-GAGA models have similar performance on the two datasets, but both are inferior to AGCRN. There are two likely causes for this result. On the one hand, STGCN requires quite high prior knowledge for designing the weighted adjacency matrix in a simple graph. Therefore, its performance largely depends on prior knowledge. On the other hand, FC-GAGA discards the prior knowledge of graphics and applies the learnable fully connected hard graph gating mechanism instead. Thus, the performance of FC-GAGA may be different on different datasets. While AGCRN uses a data-adaptive graph generation (DAGG) module to infer the inter-dependencies among different traffic series automatically, combining the advantages of the two articles intelligently. However, AGCRN still utilizes simple graphs to embed and express spatial features, allowing us to explore the validity of the hypergraph.

• The performance of our proposed model in short-term prediction is better than all baselines, indicating the effectiveness of our proposed model. Simultaneously, all the variants are inferior to the proposed model, denoting that each module is effective. The performance of the LST-HCN is better than the other two variants, the reason can be attributed to the self-learning attention matrix, playing a vital role in learning the mapping relationship between hyperedges and traffic nodes. To conclude, these experimental results suggest that the hypergraph and line graph affect capturing the high-order spatial relationship between nodes and the relationship between hyperedges. The combination of the two can fully unify complex local correlation and across road-network isomorphism in the spatial dimension in the traffic data.

• The proposed framework outperforms all the baseline models in all settings. Take the prediction of 15min as an example, compare with the best performance results on the PEMSBAY dataset, the proposed model achieves approximately 4.76% higher performance in terms of MAE, 14.85% higher in terms of RMSE, and 7.71% higher in terms of MAPE. On the PEMSM dataset, the proposed model achieves approximately 5.1% higher performance in terms of MAE, 10.21% higher in terms of RMSE, and 8.63% higher in terms of MAPE.

Parameter sensitivity analysis

In the construction of a hypergraph, a key step needs to determine the K value in the K-means clustering algorithm. An important consensus is that the K value in the K-means algorithm cannot exceed the number of road nodes. Take the prediction time step of 15min as an example, the experimental results are shown in Table 3. For the PEMSM dataset and PEMSBAY dataset, this article chooses the number of K from (4, 8, 16, 32, 64, 128) and (4, 8, 16, 32, 64, 128, 256) respectively to analyze the change of prediction precision. And the results are shown in Figs. 5A and 5B. For the two datasets, as the K value increases, the prediction precision increases, which means the effectiveness of our ideas. Nevertheless, when the inflection point appears, as the K value continues to increase, the prediction accuracy decreases instead. One possible explanation is that too many hyperedges will gradually transform the hypergraph into a simple graph. Take the most intuitive example, this article chooses the K value as the number of road nodes. Under this condition, each node is included by a hyperedge. And then the proposed framework uses the physical connection of the road network to expand the hyperedge, finally getting a simple graph. Therefore, this article chooses the K = 16 and K = 128 for PEMSM and PEMSBAY datasets, respectively. It is worth noting that for the PEMSBAY dataset when K = 256 the evaluation index of MAE is lower than when K = 128, but both RMSE and MAPE are higher than when K = 128. Our understanding is that MAE, like RMSE, measures the absolute size of the deviation between the actual value and the predicted value, while MAPE measures the relative size of the deviation. MAPE is more able to measure the stability of the model. Thus, the model can achieve the optimal effect on the PEMSBAY dataset when K = 128.

Table 3 Results of different K value on the datasets PEMSBAY and PEMSM.

Methods	PEMSBAY (15 min)	PEMSM (15 min)	
	MAE	MAPE (%)	RMSE	MAE	MAPE (%)	RMSE	
K = 4	1.420	3.351	2.651	1.880	4.408	3.257	
K = 8	1.419	3.327	2.623	1.868	4.396	3.241	
K = 16	1.408	3.276	2.605	1.863	4.337	3.229	
K = 32	1.414	3.291	2.600	1.886	4.430	3.272	
K = 64	1.409	3.283	2.597	1.883	4.381	3.253	
K = 128	1.403	3.230	2.582	1.875	4.367	3.239	
K = 256	1.403	3.262	2.591	− −	− −	− −	
Notes.

− − Denotes that the experimental results cannot be obtained.

Figure 5 The change of prediction results with different K on the two datasets.

(A) PEMSM dataset. (B) PEMSBAY dataset.

Ablation experiments

Convolutional neural networks can extract features from local spatial regions with a fixed convolution kernel size. However, for graphs, the number of other nodes connected to graph nodes is uncertain, and fixed convolution kernels cannot be used to extract features. The concept of the product began to be raised. The graph convolutional network aims to directly perform convolution on the graph and aggregate the surrounding adjacent node information to form a new node representation. To verify the effectiveness of various graph convolutions in extracting spatial features from traffic data, this article designs a series of ablation experiments.

To prove the effect of dual-channel hypergraph convolution, This article designed three ablation experiments. The training processes of the three ablation experiments are shown in Fig. 6. As the number of training epochs increases, the training set loss and validation set loss continue to decrease, indicating that the model is learning spatial–temporal features from the data without any over-fitting or under-fitting phenomenon. As is illustrated in the figures, what can be found is that the convergence process of HST-HCN and GST-HCN is slower, while the convergence processes of LST-HCN and ST-HCN are faster. The results may be attributed to the addition of the line graph, enabling us to model the spatial relationship between multiple super-points quickly. Then, the complex spatial relationships are extracted through line graph convolution, speeding up the process of expressing of spatial features. Although simple graphs and hypergraphs can model spatial connections, they cannot simultaneously analyze the features between hyper points, and the many-to-many spatial relationship in traffic data cannot be described, resulting in slow feature extraction. Therefore, it is necessary to use two-channel hypergraph convolution.

Figure 6 (A–B) Training and validation processes on the PEMSM dataset. (C–D) Training and validation processes on the PEMSBAY dataset.

At the same time, the prediction effects of all variants are shown in Fig. 7. From the histogram, what can be seen is that the values of the three evaluation indicators of ST-HCN are lower than those of the other three models, which fully demonstrates the effectiveness of the dual-channel convolution. But for the two datasets, there are some subtle differences. For the PEMSM dataset, simple graph convolution can also achieve good results, because when designing the adjacency matrix with weights in this dataset, a large number of traffic nodes are screened to ensure that the maximum eigenvalue of the adjacency matrix is about 2. By doing so, the high effectiveness of layer-wised graph convolution can be guaranteed. For the two datasets of PEMSM and PEMSBAY, the effect of only using hypergraph convolution is not obvious. That is because this article need to ensure that the model learns the relationship between super edges and nodes while extracting the high-order spatial relationships between nodes. Only by doing this, the proposed framework can fully discover the complex local correlation and across road-network isomorphism in the spatial dimension hidden in the traffic data. Both variants of the proposed method do not perform as well as ST-HCN, which indicates the effectiveness of dual-channel hypergraph convolution.

Figure 7 Result change with different convolution module.

(A) PEMSM dataset. (B) PEMSBAY dataset.

From the results of the above ablation experiments, we can see the necessity of using a space–time hypergraph. The construction process of the hypergraph discovers important traffic nodes globally and uses spatial connections to complete them locally so that the complex spatial characteristics of traffic data can be integrated. In addition, for hypergraphs, multiple points can be directly placed on an edge for feature extraction, which is also of great significance for modeling the spatial isomorphism across road networks in traffic data.

To demonstrate the effectiveness of the proposed hypergraph construction method, this article designed two models: GST-HCN and HST-HCN. GST-HCN uses simple graphs for training, while HST-HCN applies hypergraphs for training. The experiment shows that HST-HCN performs better. Moreover, to demonstrate the effectiveness of dual-channel hypergraph convolution, this article designed two models: LST-HCN and ST-HCN. LST-HCN utilizes line graphs for training, while ST-HCN uses dual-channel hypergraph convolution for training. The experiment shows that ST-HCN performs better. In summary, the results of the ablation experiment could support the main contributions of the article.

Case study

To better illustrate the learning ability and prediction effect of the model, this article visualizes the results of different training epochs. Taking the performance results of the model on the PEMSBAY dataset as an example, a cluster center in the K-means clustering algorithm is selected to display the effect. As is shown in Fig. 8, from the changing trend of the blue solid line, what can be seen is that the average speed of this traffic node is maintained at a high level before 6:00 in the morning, from 10:00 in the morning to 2:00 p.m. and after 8:00 in the evening. The activity trajectory of several periods is not obvious. While at around 8:00 in the morning and around 5:00 p.m., the speed of traffic nodes is significantly reduced. This is because most people choose to drive to and from get off work at these times, resulting in a rapid increase in road occupancy and traffic congestion. Therefore, the traffic speed at key intersections has obvious periodicity. This fully shows that the K-means algorithm can effectively select the key traffic intersections from the road network, and then can capture the high-order spatial relationship between key traffic intersections and other traffic nodes through hypergraph convolution.

Figure 8 Visualization of prediction results on PEMSBAY dataset.

The solid blue line represents the true value of the traffic node, and the orange dotted line represents the predicted value of the model. (A) Epoch 1, (B) Epoch 15, (C) Epoch 30, (D) Epoch 50.

After one cycle of model training, the prediction results of the model have been able to track the changing trend of the true value well, which shows that our proposed model can converge quickly and achieve a relatively good result. Meanwhile, the results also show from the side that the dual-channel convolution is very effective to extract the spatial features of traffic data. Furthermore, the proposed framework used LSTM with a convolution module to extract the temporal features of traffic data, which can quickly represent the data distribution so that the satisfactory results can be obtained after one cycle of training. After that, according to the overall distribution characteristics of the data, the model gradually learns, and the fitting effect gradually increases. Even in a period with obvious fluctuations, it can track very well. This indicates that our proposed model can learn the nonlinear features in the traffic data well and cope with the dynamic changes of the traffic data, thereby realizing the modeling of the traffic data in a completely data-driven manner.

Analysis of research results and discussion

The above experimental results fully demonstrate the effectiveness of our proposed framework. However, from the experimental results table, it can be found that the effect of model improvement is gradually insignificant. Therefore, to further analyze the advantages and disadvantages of the model, this article compares the best-performing model AGCRN in the dataset with our proposed model.

Taking the two main evaluation indicators MAE and MAPE as examples, this article visualizes the prediction results. As is illustrated in Fig. 9, for both datasets, as the prediction time step increases, the values of the two model evaluation indicators rise steadily, indicating that the prediction effect of the model decreases with the increase of the time step. However, what can be seen is that the distance between the two model evaluation indicators is getting smaller and smaller, which shows that although our model has achieved good results in the time of short-term traffic forecasting, as the forecasting time increases, AGCRN has beyond the trend. This is largely because AGCRN uses a data-adaptive graph generation module to infer the inter-dependencies among different traffic time series automatically. Our model does not add this long-term dependency, allowing the model to infer traffic information for longer temporal distances. On the one hand, this fact shows that our model is not perfect, but it points us to the next steps in our work. On the other hand, this article focuses on exploring the effectiveness of hypergraphs in modeling traffic data and make full use of hypergraphs to solve the problem of the across road-network isomorphism in traffic prediction. The experimental results also fully demonstrate up to this point. Therefore, our work is still of great interest in the field of how to use hypergraphs to model traffic data.

Figure 9 Results change with different methods on different datasets.

(A–B) PEMSM dataset. (C–D) PEMSBAY dataset.

This work aims to provide accurate forecasting information by analyzing historical traffic data. Therefore, the managerial implications of this work mainly have two aspects. One is that accurate forecasting information can help traffic managers understand road conditions better. The other is that the operating status of the vehicles in the network can help urban residents optimize their travel experience.

Conclusion and Future Work

In this article, a spatial–temporal hypergraph convolutional network for traffic forecasting was proposed. The dual-channel graph convolutional neural network was proposed, which can effectively capture the many-to-many spatial relationship between traffic nodes and can perform feature extraction on the relationship between hyper-points. Simultaneously, to tackle the periodic drift problem in the temporal dimension, an LSTM network with a convolution module was applied. Moreover, experiments demonstrate that ST-HCN outperforms the existing methods in short-term traffic forecasting.

In the future, our work will consider long-term dependencies between time steps to achieve effective results in mid-to-long-term traffic forecasting. As mentioned in the article, the proposed model performs spatiotemporal modeling of traffic data in a data-driven manner, so the model can also be applied to other spatiotemporal graph modeling and forecasting tasks, such as urban demand forecasting, urban pedestrian flow forecasting, and so on.

Supplemental Information

Supplemental Information 1 All relevant files of data and code

PEMSBAY contains the traffic speed on 325 road sensors in the Bay Area. In addition, the readings of all sensors deployed on the road are also aggregated into 5-minutes windows. It can be obtained from the URL: https://github.com/Azusa-Yuan/fc-gaga-torch/tree/main/datasets. PEMSM contains the traffic speed on 228 road sensors in the District 7 of California. It can be obtained from the URL: https://github.com/VeritasYin/STGCN_IJCAI-18/tree/master/dataset

Each dataset can be obtained from the provided URL. Taking the PEMSBAY dataset as an example, we provide the executable source code, including the construction of hypergraph and the overall training process of hypergraph convolution neural network.

Click here for additional data file.

Additional Information and Declarations

Competing Interests

Author Contributions

Data Availability

Xiangjie Kong is an Academic Editor for PeerJ. Junchen Jin is employed by Zhejiiang Supcon Information CO., LTD. The authors declare there are no competing interests.

Zhenzhen Zhao conceived and designed the experiments, performed the experiments, analyzed the data, performed the computation work, prepared figures and/or tables, authored or reviewed drafts of the article, and approved the final draft.

Guojiang Shen conceived and designed the experiments, analyzed the data, authored or reviewed drafts of the article, and approved the final draft.

Junjie Zhou performed the experiments, analyzed the data, performed the computation work, prepared figures and/or tables, and approved the final draft.

Junchen Jin performed the experiments, analyzed the data, performed the computation work, prepared figures and/or tables, and approved the final draft.

Xiangjie Kong conceived and designed the experiments, analyzed the data, authored or reviewed drafts of the article, and approved the final draft.

The following information was supplied regarding data availability:

The raw data is available in the Supplemental Files.

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
