# Peer review of "Spatial-temporal hypergraph convolutional network for traffic forecasting"

_PeerJ Computer Science, doi:10.7717/peerj-cs.1450_

## Round 0.1 · original submission · Major Revisions

Dear authors,

Your paper has been reviewed by three reviewers who suggested revisions. Please make appropriate changes and write cover letter with replies to reviewers point to point.

Reviewer 1 ·

Basic reporting

no comment'

Experimental design

no comment'

Validity of the findings

no comment

Additional comments

The experiment shows very well the rationality of the proposed model. A good example of spatiotemporal modeling of traffic data and the multiple possibilities of application

Cite this review as

Reviewer 2 ·

Basic reporting

Special praise to the author for the methodology of the paper and the given case study. To make the work even better, I give certain suggestions:
In the abstract (18th, 19th, 21th, 22th line), as well as in other parts of the paper, the first person plural is used to write the paper (35th,55th,63th,64th, 67th, 68th 69th 71th 72th 76th 80th 84th 86th 87th 88th 89th 92th 94th 169th 172th 173th 194th 198th 200th 201th 209th 214th 216th 217th 222th 224th 226th 228th 230th 233th 237th 238th 249th, After Algorithm1, 255th 261th 265th, After 268th , After Figure 4, 276th 280th 284th 287th 290th 296th 298th 305th 307th 309th 312th 314th 341th 342th 385th 391th 392th 393th 398th 401th 405th 414th 421th 422th 427th 429th 430th 441th 452th 454th 457th 464th 466th 469th 477th). That needs to be corrected. The paper can not be written in the first person!

Suggestion to authors: Instead of the title "2. Related Works" use "LITERARY REVIEW"....Not mandatory.
Also, chapter 2 needs to add an INTRODUCTION section (from line 39 to line 54).
Algorithm 1: Is it possible to show this as an algorithm (rather than a program)? Make a figure.
Line 284-289- Must state what speed it is? State that it was obtained by local measurement
Line 296 - State which 35 days were used in the research?
Images should follow the chronology of the text in the paper (For example figure 6, 7, 8, Line 430 ……in Figure 9…..- page 13. Figure 9 is on page 15!)
In the discussion, a picture is given which is found in the conclusion. This part of the text must be adjusted and rearranged.
499 - Instead of a discussion chapter, I suggest to the authors that the chapter be titled ANALYSIS OF RESEARCH RESULTS AND DISCUSSION
At the end of the discussion, add Figure 10. and comment this Figure. Remove figure 10 from the conclusion!!
In the conclusion, a concrete result of the research must be given!! (I don't see it...)

Experimental design

No Comment

Validity of the findings

It is a very interesting research, based on an intelligent transportation systems... The paper represents an extremely important segment of research in road engineering, but it is written unsystematically in some parts.
Figure 2 is very well shown. Also, the meaning per section is clearly explained!

Additional comments

No comment

Cite this review as

Reviewer 3 ·

Basic reporting

The following concerns are raised after reading the paper:
The related literature reviewed should be followed by a comprehensive research gap analysis;

Experimental design

The main contribution is not well described and justified;
The convolutional network concept needs to be more elaborated with respect to different processes given in the literature;
The significance of using integrated Spatial-Temporal hypergraph in comparison with other techniques already worked out in the literature is not well explained;

Validity of the findings

What is exactly new in the mathematical formulations should be pointed out;
The numerical study needs more elaboration and analysis using more multi-dimension illustrations;
Tables and figures need to be explained;
The managerial implications should be well discussed in the application areas of the approach.

Cite this review as

---

## Round 0.2 · accepted · Accept

All reviewers have accepted the paper.

Reviewer 2 ·

Basic reporting

no comment

Experimental design

no comment

Validity of the findings

no comment

Additional comments

No comment

Cite this review as

Reviewer 3 ·

Basic reporting

Past comments are fulfilled.

Experimental design

Past comments are fulfilled.

Validity of the findings

Past comments are fulfilled.

Additional comments

Past comments are fulfilled.

Cite this review as